# A Soft Zwitterionic Hydrogel as Potential Coating on a Polyimide Surface to Reduce Foreign Body Reaction to Intraneural Electrodes

**DOI:** 10.3390/molecules27103126

**Published:** 2022-05-13

**Authors:** Manuele Gori, Sara Maria Giannitelli, Gianluca Vadalà, Rocco Papalia, Loredana Zollo, Massimo Sanchez, Marcella Trombetta, Alberto Rainer, Giovanni Di Pino, Vincenzo Denaro

**Affiliations:** 1Institute of Biochemistry and Cell Biology (IBBC)—National Research Council (CNR), International Campus “A. Buzzati-Traverso”, Via E. Ramarini 32, 00015 Rome, Italy; 2Laboratory of Regenerative Orthopaedic, Research Unit of Orthopaedic Surgery, Campus Bio-Medico University of Rome, 00128 Rome, Italy; g.vadala@policlinicocampus.it (G.V.); r.papalia@policlinicocampus.it (R.P.); denaro@policlinicocampus.it (V.D.); 3Laboratory of Tissue Engineering and Chemistry for Engineering, Department of Engineering, Campus Bio-Medico University of Rome, 00128 Rome, Italy; s.giannitelli@unicampus.it (S.M.G.); m.trombetta@unicampus.it (M.T.); a.rainer@unicampus.it (A.R.); 4Department of Orthopaedic and Traumatology, Fondazione Policlinico Universitario Campus Bio-Medico, 00128 Rome, Italy; 5Research Unit of Advanced Robotics and Human-Centred Technologies, Department of Engineering, Campus Bio-Medico University of Rome, 00128 Rome, Italy; l.zollo@unicampus.it; 6Core Facilities, Istituto Superiore di Sanità (ISS), 00161 Rome, Italy; massimo.sanchez@iss.it; 7Institute of Nanotechnology (NANOTEC)—National Research Council (CNR), 73100 Lecce, Italy; 8NeXT: Neurophysiology and Neuroengineering of Human-Technology Interaction Research Unit, Campus Bio-Medico University of Rome, 00128 Rome, Italy; g.dipino@unicampus.it

**Keywords:** intraneural electrodes, Polyimide, PEG, poly(SBMA), zwitterionic hydrogel, surface coating, foreign body reaction, myofibroblasts, macrophages, Everolimus

## Abstract

Invasive intraneural electrodes can control advanced neural-interfaced prostheses in human amputees. Nevertheless, in chronic implants, the progressive formation of a fibrotic capsule can gradually isolate the electrode surface from the surrounding tissue leading to loss of functionality. This is due to a nonspecific inflammatory response called foreign-body reaction (FBR). The commonly used poly(ethylene glycol) (PEG)-based low-fouling coatings of implantable devices can be easily encapsulated and are susceptible to oxidative damage in long-term in vivo applications. Recently, sulfobetaine-based zwitterionic hydrogels have emerged as an important class of robust ultra-low fouling biomaterials, holding great potential to mitigate FBR. The aim of this proof-of-principle in vitro work was to assess whether the organic zwitterionic—poly(sulfobetaine methacrylate) [poly(SBMA)]—hydrogel could be a suitable coating for Polyimide (PI)-based intraneural electrodes to reduce FBR. We first synthesized and analyzed the hydrogel through a mechanical characterization (i.e., Young’s modulus). Then, we demonstrated reduced adhesion and activation of fibrogenic and pro-inflammatory cells (i.e., human myofibroblasts and macrophages) on the hydrogel compared with PEG-coated and polystyrene surfaces using cell viability assays, confocal fluorescence microscopy and high-content analysis of oxidative stress production. Interestingly, we successfully coated PI surfaces with a thin film of the hydrogel through covalent bond and demonstrated its high hydrophilicity via water contact angle measurement. Importantly, we showed the long-term release of an anti-fibrotic drug (i.e., Everolimus) from the hydrogel. Because of the low stiffness, biocompatibility, high hydration and ultra-low fouling characteristics, our zwitterionic hydrogel could be envisioned as long-term diffusion-based delivery system for slow and controlled anti-inflammatory and anti-fibrotic drug release in vivo.

## 1. Introduction

Invasive neural interfaces, implanted within peripheral nerve stumps of amputees, allow biomedical engineers to artificially interface the nervous system to the external environment. A primary aim, among others, is to bidirectionally control neuroprosthetic devices [1]. However, intraneural localization of such interfaces implies a lack of long-lasting stability, with a consequent reduction of performance over time, largely due to the reaction that the body mounts around them [2,3]. Indeed, our bodies naturally tend to insulate and exclude any implanted foreign material from the surrounding environment through a physiological response known as foreign body reaction (FBR) [2,4]. This response, which is carried out by the innate immune system, is characterized by a cascade of cellular events that shares many aspects of wound healing [5], starting with plasma protein adsorption, e.g., fibrinogen, complement and antibodies. It elicits coagulation around the implant and adhesion of circulating leukocytes and monocytes to the surface, leading to the differentiation of the latter cells into activated M_1_ macrophages [3,6]. The M_1_ macrophages are pivotal players in the crosstalk with other inflammatory and pro-fibrotic cells, such as myofibroblasts, recruited and activated by pro-inflammatory cytokines (e.g., TGF-β) further produced by M_1_ macrophages. Finally, in the last stage of the process, myofibroblasts are responsible for the encapsulation of the implant by secreting extracellular matrix (ECM) components that tend to isolate the intraneural device, compromising its long-term functionality [4]. Notably, surface chemistry and material characteristics as well as hydrophobicity, strongly influence the composition of the adsorbed molecules and the consequent cellular cascade of FBR [7], thus underlining the importance of the surrounding biomaterial coating around typical Polyimide (PI)-based implants that could impact the biocompatibility and safety of the implanted interface [3,4,8,9,10]. In this regard, poly(ethylene glycol) (PEG) [11,12,13] and poly(2-hydroxyethyl methacrylate) (pHEMA) [14,15], which have long been the most common coatings used to overcome the undesired FBR to PI-based implantable interfaces, were recently replaced by the more stable poly(carboxybetaines), poly(phosphobetaines) and poly(sulfobetaines) zwitterionic hydrogels [16,17,18,19]. These hydrogels present advantageous features in terms of hydrophilicity, biocompatibility and anti-inflammatory and ultra-low fouling characteristics, both in vitro and in vivo [20,21,22]. As a matter of fact, PEG and pHEMA-based coatings are highly susceptible to oxidative damage, severe immune response and fibrotic encapsulation over time [13,23,24], limiting their extended stability in a biological environment. On the contrary, zwitterionic hydrogels are able to modulate macrophage polarization towards anti-inflammatory phenotypes in vitro and in animal models [20,21,22]. They also possess high intrinsic ionic conductivity, a useful property in the implantation of flexible electrical devices [25]. Despite significant interest in developing novel polymeric zwitterionic coatings of medical devices, little has been done to date to investigate the possibility of photochemically coupling zwitterionic thin film polymers made of poly(sulfobetaine methacrylate) [poly(SBMA)] to PI surfaces. To our knowledge, no evidence currently exists regarding the use of such hydrogels as functional coatings to protect intraneural interfaces from FBR.

Therefore, the main purposes of the present work were: (i) to assess the advantageous features of our soft poly(SBMA) zwitterionic hydrogel in reducing FBR, (ii) to increase the protective effects of the hydrogel against FBR through the prolonged release of a model drug and (iii) to fabricate a stable coating of a PI surface with a thin film of the zwitterionic hydrogel. Importantly, we confirmed the high biocompatibility, ultra-low fouling properties of the poly(SBMA) zwitterionic hydrogel in vitro, as well as low activation of human myofibroblasts and macrophages, compared with PEG and polystyrene control surfaces, through confocal microscopy-based high-content analysis (HCA). We also carried out a photo-immobilization of a thin film of the poly(SBMA) zwitterionic hydrogel on a PI surface through a covalent bond. Finally, as an interesting novelty, we showed the sustained release of a model anti-fibrotic compound from the hydrogel. Thereby, we demonstrated the potential capacity of our hydrogel as a coating material for invasive neural interfaces, for long-term release of anti-inflammatory and anti-fibrotic drugs in future in vivo applications against FBR.

## 2. Results

### 2.1. Functionalization of Polyimide Surface with PEG Block Copolymer

A novel coating procedure to functionalize the surface of a layer of PI (aka Kapton) with a 2-[Methoxy(polyethyleneoxy)9-12propyl]trimethoxysilane was developed. The bonding process of the PEG coating with the PI surface lay on the PI exposure of the hydroxyl groups, generated by the ionized O_2_ plasma treatment, which interacted with the nearly free silanol surface moieties of PEG thereby forming a stable chemical bond (Figure 1A). In more detail: following hydrolysis of methoxy silane groups, PEG-silane was grafted to a plasma-treated Kapton surface via condensation of formed silanols with surface hydroxyl groups in methanol. Further stabilization occurred via siloxane bonds between immobilized PEG-silane chains. The hydrophilicity of the obtained PEG surface was then tested by measuring the static contact angle of a water drop (53.22 ± 1.87°, Figure 1A). This result was in line with data from the literature [26], thereby confirming the higher hydrophilicity of the PEG-coated surface vs. the more hydrophobic pristine PI surface (70.76 ± 1.66°).

### 2.2. Biocompatibility of PEG Coating

The highly resistant and biocompatible polymer PI is often the lective material used for the core of flexible neural interfaces, taking advantage of its high fabrication versatility and applicability [27,28,29]. Most of the polymer coatings of intraneural PI electrodes used in the literature were made of PEG or PEG-based hydrogels that exploited their high biocompatibility and low-fouling features [30,31,32]. However, their limitations, e.g., their intrinsic physico-chemical properties, can give rise to oxidative stress processes, limiting their long-term performance [33,34]. In this study, to use PEG coating as a standard in cell culture experiments, activated myofibroblasts (representative immunofluorescence confocal image in Figure 1B) were seeded at a density of 2.5 × 10^3^/cm^2^ and cultured on top of a PI surface vs. PEG-coated PI surface stretched over CellCrown inserts (experimental scheme in Figure 1A). After 24 h in culture, we verified PEG biocompatibility via Vybrant cytotoxicity assay as previously described [35]. Comparable (with no statistical difference, *p* = 0.93) and very high levels of cell viability (Figure 1C), close to 95%, were detected between the nude PI surface and PEG-coated PI surface (i.e., 94.23 ± 1.04% vs. 94.03 ± 1.73%, respectively) confirming the excellent biocompatibility of the PEG polymer observed in the literature [30,32,36]. Furthermore, confocal fluorescence images of activated myofibroblasts [stained in green with α-Smooth Muscle Actin (α-SMA) antibody in Figure 1D] showed a statistically highly significant reduction in adhesion after 24 h on PEG-coated PI vs. PI surfaces (Figure 1E) both with (*p* < 0.0001) and without (*p* < 0.0001) human plasma fibrinogen (at 4.5 mg/mL, which is in the range of the highest concentrations circulating in human plasma, see experimental scheme in Figure 1F) [37,38] to mimic, to some extent, a physiological-like condition with plasma protein adsorption on the surfaces [39,40]. Accordingly, living myofibroblasts showed a completely different morphology between the two surfaces (Figure 1D), with a few round cells on PEG-coated PI. That is the typical shape of low-/non-activated cells, as a consequence of poor adhesion [11,22], compared to those on pristine PI, which spread well over the whole surface, increasing their adhesion area with no clusters in keeping with results from [12,22].

### 2.3. Synthesis and Characterization of the Poly(SBMA) Zwitterionic Hydrogel vs. Polystyrene Control Surfaces

The poly(SBMA) zwitterionic hydrogel was successfully synthesized slightly modifying the protocol developed by [21,41]. Then, it was characterized in terms of mechanical behavior (Figure 2). The Young’s modulus was 2.7 ± 0.24 kPa, which reflected that of the neural tissue (i.e., between 0.1 and 10 kPa) [42] minimizing the mechanical mismatch between implanted devices and tissue, which could give rise to undesired host responses, exacerbating the inflammatory state. Although the PI itself reduces the mismatch existing between the implant and surrounding tissue, yet its Young’s modulus is higher than that of neural tissue [42], underlining the important role played by substrate stiffness on fibrotic reactions. It is well known that the mechanical mismatch represents one of the main causes of adverse tissue reactions to an implant [43,44]. This useful hydrogel property, which can vary depending on the PEGDMA crosslinker concentration used [41], could thus be harnessed to contrast fibroblast and macrophage activation, as well as pro-inflammatory response: the softer the hydrogel, the lower the elicited FBR [11]. As a further characterization, a biocompatibility analysis was performed for the zwitterionic hydrogel, using Vybrant cytotoxicity assay on hydrogel-derived supernatant compared with polystyrene control-derived supernatant, to verify the effects of soluble substances extracted from the hydrogel on activated myofibroblasts after 24 h in culture on a 24-well tissue culture plate (Figure 3A). The latter showed a cell viability close to 100% (i.e., 96.7 ± 0.60% in hydrogel supernatant) with no significant difference (*p* = 0.072) vs. cells grown in control-derived supernatant (i.e., 99.51 ± 0.12%). Furthermore, activated myofibroblasts were seeded at a density of 2.5 × 10^4^/cm^2^ [22] on a polystyrene surface and a hydrogel surface. They were then cultured for 24 h and 48 h for cell viability analysis (Figure 3B). Very high (and comparable) levels of cell survival, with no statistical differences (*p* > 0.05), were detected after 24 h and 48 h between the polystyrene (i.e., 97.80 ± 1.045% and 96.71 ± 1.76%, respectively) and hydrogel surfaces (i.e., 93.15 ± 1.05% and 93.01 ± 1.17%, respectively). This was similar to our previous data for the PEG-coated PI surface after 24 h (Figure 1C). Collectively, our zwitterionic hydrogels showed almost no cytotoxicity up to 48 h, in agreement with other works [20,22,41,45,46]. Finally, Live/Dead cell viability assays were carried out at day 1 and day 4 after myofibroblast seeding (Figure 3C), showing very low percentages of dead cells (in red) after 1 and 4 days, with no statistically significant differences between the two surfaces (*p* = 0.29 and *p* = 0.43, respectively). This confirmed the results shown in Figure 3B after 24 h and 48 h. Interestingly, cell morphology on the hydrogel, similar to that observed on PEG-coated PI (Figure 1D), was characterized by a few scattered cells with round shapes, rather than the well spread and semiconfluent myofibroblasts seen on the polystyrene control plates (Figure 3D). Taken together, these results clearly demonstrated a low capacity for adhesion of myofibroblasts in long-term cultures on the zwitterionic polymer, which could help reduce myofibroblast activation and their pivotal contribution to the progression of FBR processes.

In addition to the cell viability assays, a comparison of cell adhesion was also performed (Figure 3E) between myofibroblasts (stained in green with CMFDA dye and nuclei in blue with DAPI in Figure 3F) seeded on hydrogel and polystyrene surfaces and cultured for 24 h, with and without human plasma fibrinogen (at 4.5 mg/mL), as shown in Figure 1D,E. The number of adherent cells on the hydrogel was much lower than polystyrene in a statistically significant manner, with and without fibrinogen. Moreover, cell morphology between the two surfaces was totally different, in agreement with what we observed earlier in the Live/Dead assay (Figure 3D). Lastly, a representative micrograph of the interface between culture medium and hydrogel surface (which was cut along a sagittal plane and overturned) showed myofibroblasts growing on the surface but not in the thickness of the gel (Figure 3G), thereby implying no cell invasion of the gel, with possible consequent detrimental effects on the implant surface. Thus, although the cell viability analyses performed in Figure 3A–C showed comparable levels of cell survival between the two surfaces, the number of myofibroblasts capable of adhering to the hydrogel was dramatically lower than on polystyrene, with the typical appearance of non-activated cells. Two main characteristics, known from the literature, could have contributed to the observed low adherence and low activation of pro-fibrotic cells: the low stiffness of the hydrogel, as observed above, and its non-biofouling properties, attributable to the high surface hydration of zwitterionic chemistries hindering hydrophobic interactions with cell membranes and cell adhesion proteins [16,21,47,48,49,50,51]. Such advantageous behavior could help overcome FBR over time, thus making this zwitterionic hydrogel a promising solution.

### 2.4. Human Macrophage Activation on Zwitterionic Hydrogel vs. Polystyrene Control Surfaces

We also set out to investigate the behavior of human macrophages on the hydrogel surface, as their attachment and activation is known to be critical for the extent of the chronic inflammatory response occurring during FBR [52,53]. To study their activation, we used the human monocytic cell line THP-1. Upon differentiating into macrophages (differentiation of the M_0_ and M_1_ phenotypes was assessed by measuring the expression of the cell surface and intracellular markers CD86 and CD68 (a pan-macrophage marker) by flow cytometry [54,55,56] (as shown in Appendix A) cell morphology, following adhesion, and oxidative stress induction (i.e., generation of Reactive Oxygen Species (ROS) and Reactive Nitrogen Species (RNS), including nitric oxide (NO) [21,22]) were analyzed through high-content screening of fluorescence confocal images after 24 h in culture on zwitterionic surface vs. tissue culture polystyrene surface (Figure 4). Once differentiated to M_0_ macrophages, using PMA for 48 h, 2.5 × 10^4^/cm^2^ THP-1 cells were cultured onto the two surfaces for 24 h with and without stimulation with lipopolysaccharide (LPS) and human recombinant gamma-interferon (IFN-γ), inducing their polarization to M_1_ pro-inflammatory macrophages [54,55,57].

A statistically significant difference in macrophage-induced oxidative stress (green fluorescence in Figure 4B) was found between non-stimulated and stimulated cells on polystyrene (Figure 4A, *p* = 0.049), thereby showing the effects of LPS + IFN-γ-induced polarization on macrophages used as a positive control for inflammation. No significant differences were detected between the same conditions on the hydrogel surface (Figure 4A). Interestingly, a statistically significantly higher ROS/RNS generation (green fluorescence in Figure 4B) was observed in stimulated cells grown on polystyrene compared to those on hydrogel (Figure 4A, *p* = 0.0056 and *p* = 0.029 vs. non-stimulated and stimulated, respectively), thereby confirming the ability of the zwitterionic surface in inhibiting M_1_ activation, and in turn reducing the associated inflammatory state [53,58]. The ROS/RNS inducer Pyocyanin was used as a further positive control of oxidative stress, according to manufacturer’s instructions (ENZO Life Sciences, Farmingdale, New York, NY, USA), for both surfaces with high inducing effects. Furthermore, the morphology of macrophages showed a different behavior on the hydrogel, with small clusters of round cells, growing in scattered areas as in [22], compared to many isolated cells on polystyrene (Figure 4B). Thus, the overall macrophage inflammatory response on the hydrogel surface was lower than polystyrene controls both with and without stimulation with the specific inducers LPS + IFN-γ. Their behavior in such a context was the result of a complex mechanism that depended on surface chemistry and Young’s modulus [22,53,59].

### 2.5. Photoimmobilization of the Poly(SBMA) Zwitterionic Hydrogel on Polyimide Surface and Adhesion Test

One of the primary aims of this work was the identification of the best method for a covalent attachment, through a UV-light initiated free radical polymerization of a thin film of the soft zwitterionic hydrogel (with a thickness in the micrometer range) to the PI surface. To the best of our knowledge, a PI surface has never been photochemically coated with a thin hydrogel film made of a poly(SBMA) chemistry to explore its potential use against FBR to an implantable neural interface. Therefore, we modified the existing methodology for the grafting of zwitterionic hydrogel onto hydrophobic membranes via O_2_ plasma and photochemical treatment [60] (Figure 5). This turned out to be the most suitable method, as verified by the analysis of myofibroblast adhesion (Figure 6B,C). In detail, the first step of the O_2_ plasma treatment activated the surface of the PI membrane, leading to the emergence of new functional groups. Under UV irradiation treatment, these functional groups were unstable and were thereby able to bind the carbon atom of the SBMA, grafting the SBMA molecules onto the membrane. At the same time, the polymerization of the SBMA arose from the breaking of the carbon-carbon double bond of the SBMA monomers and PEGDMA crosslinker. Interestingly, after 24 h in culture on a layer of PI (stretched over a CellCrown insert as shown in Figure 6A) which was functionalized with thin films of the polymer through the methods detailed in Figure 6B (i.e., direct deposition of the thin film vs. chemical grafting on the surface), the lowest number of adherent myofibroblasts was observed on the O_2_ plasma-treated surface as compared to nude PI (*p* < 0.0001) in Figure 6B, which showed a round shape. Afterward, wettability of the polyzwitterionic surface was measured through analysis of the static contact angle of a water drop, in order to confirm the role played by high hydration in reducing cell adhesion (Figure 6D). The hydrogel layer modified the PI wettability from a value of 70°, which was typical of hydrophobic surfaces and in agreement with the literature [22,61], to the statistically significantly lower value of 30° (*p* < 0.0001), which was in the range observed for other hydrophilic zwitterionic layers [22,62,63,64]. Surface hydration is one of the key elements responsible for zwitterionic ultra-low fouling properties, as it forms a water layer (via ionic solvation) that represents a physical and energetic barrier to unwanted protein adsorption and cell attachment to the surface [65,66]. While pHEMA and PEG hydrogels, currently used in many applications [30,67,68], have lower hydration than native tissue, and their functionalization using hydroxyl groups is quite difficult to obtain [24], solutions based on zwitterionic hydrogels such as the one presented herein are more convenient and versatile, and could overcome the aforementioned limitations.

### 2.6. Proof-of-Principle Test of the Release of an Anti-Fibrotic Drug Compound from the Zwitterionic Hydrogel

In order to locally modulate the innate immune response of the host tissue to the implant, our hydrogel formulation could represent a useful tool to encapsulate possible therapeutic anti-inflammatory and anti-fibrotic compounds, as similarly described in other works [69,70]. The hydrogel swelling ability, observed during synthesis, could also be exploited to easily adsorb aqueous drug compounds that could be released in the surrounding milieu in a slow and controlled manner. To this aim, a list of potential therapeutic drug candidates could be loaded during polymer synthesis into our zwitterionic hydrogel coating including, among others, the anti-fibrotic and TGF-β blocking drugs Everolimus, Stat3 inhibitor (S3I-201) and various Rho-associated protein kinase (ROCK) inhibitors [71,72,73,74,75,76,77]. This tailored strategy will represent the aim of our future works, harnessing the present poly(SBMA) zwitterionic hydrogel as an effective drug delivery system in vitro as well as in animal models of FBR. TGF-β, which is a central mediator in fibrogenesis, could represent a promising target in gene therapies against pro-inflammatory and pro-fibrotic factors, as shown in different fibrotic diseases [78,79,80]. The mammalian target of rapamycin (m-TOR) inhibitor, Everolimus, has shown important anti-fibrotic effects [81], and has been used herein as proof-of-principle drug for its potential therapeutic application against FBR in vitro as well as in vivo. Thus, this model drug has been exploited for measuring the drug release capacity of the hydrogel in physiological conditions (i.e., in PBS at 37 °C in a humidified atmosphere containing 5% CO_2_). As shown in Figure 7, its release showed a gradual and continuous increase (i.e., a sustained and quite linear trend) for over a month, with almost the 50% (i.e., 48,24 ± 2.45%) of the total drug released after three weeks (i.e., 24.12 ± 1.23 μg at day 21). The initial burst release observed within the first 8 h (i.e., 20.47 ± 3.46%) could be due to the contribution of the drug adsorbed on the outer surface of the hydrogel during the loading step. These results clearly demonstrated the possibility of a controlled drug release around the implant.

Of note, different kinetics of release for different drugs were to be expected. These could be influenced by the diverse physicochemical properties of the tested compounds, including solubility, hydrophilicity, molecular weight and size. These different features could therefore determine either a higher or lower retention into the hydrogel mesh and must be taken into careful consideration for future therapeutic purposes with different drug compounds.

## 3. Materials and Methods

### 3.1. Chemicals and Cells

The 2-[Methoxy(polyethyleneoxy)9-12propyl]trimethoxysilane (Mn = 591–719, relative density = 1.071 g/mL) was purchased from Fluorochem Ltd., Glossop, UK; CellCrown 24 inserts were purchased from Scaffdex Oy (Tampere, Finland); Dulbecco’s Modified Eagle Medium (DMEM), L-glutamine and penicillin/streptomycin solution were all purchased from Lonza (Basel, Switzerland); Fetal bovine serum (FBS, Gibco), MEM nonessential amino acids solution, CellTracker Green CMFDA Dye, DAPI solution, BlockAid Blocking solution (B10710), Vybrant cytotoxicity assay (V-23111), Live/Dead Cell Imaging Kit (R37601), eBioscience fixable viability dye eFluor 780, goat anti-mouse IgG Alexa Fluor-488, goat anti-rabbit IgG Alexa Fluor-568 and Hoechst 33342 nucleic acid stain (H3570) were all purchased from Thermo Fisher Scientific (Waltham, MA, USA); lipopolysaccharides (LPS from E. coli O55:B5), Interferon- γ human recombinant (IFN- γ) and EDTA solution 10× (59418c) were purchased from Merck (Darmstadt, Germany); [2-(Methacryloyloxy)ethyl]dimethyl-(3-sulfopropyl)ammonium hydroxide (SBMA, Mn = 279.35) as monomer, poly(ethylene glycol) dimethacrylate (PEGDMA, Mn = 550, density = 1.099 g/mL) as crosslinker and Photoinitiator Irgacure 2959 (I2959, Mn = 224.25) as initiator, Phosphate Buffered Saline (PBS) without Ca^2+^ and Mg^2+^, Fibrinogen from human plasma, Paraformaldehyde (PFA), Triton X-100, Human Serum (from human male AB plasma, H4522), Tween 80, RPMI-1640 Medium, DMEM 4500 mg/L glucose (High Glucose, D0819) and Phorbol 12-myristate 13-acetate (PMA, P1585) were all purchased from Sigma-Aldrich (Merck, Darmstadt, Germany); Monoclonal Mouse Anti-Human α-Smooth Muscle Actin antibody, clone 1A4, was purchased from Agilent—DAKO (Santa Clara, CA, USA); Polyclonal Rabbit Anti-Mammalian Collagen Type I antibody was purchased from Rockland Immunochemicals Inc. (Limerick, PA, USA); Human dermal fibroblasts and THP-1 cells were purchased from ATCC (Manassas, VA, USA); FITC-conjugated monoclonal mouse anti-human CD86 antibody and PE-conjugated monoclonal mouse anti-human CD68 antibody were both purchased from ImmunoTools GmbH (Friesoythe, Germany); ROS-ID Total ROS/RNS/Superoxide Detection kit (ENZ-51010) was purchased from ENZO Life Sciences, Inc. (Farmingdale, New York, NY, USA); Everolimus (RAD001) was purchased form Selleckchem.com (Houston, TX, USA).

### 3.2. Functionalization of Polyimide (Kapton) Surface with PEG

In brief: 2-[Methoxy(polyethyleneoxy)9-12propyl]trimethoxysilane was bound to the PI surface. First, strips of PI were placed into a FEMTO plasma cleaner (Diener Electronic, Ebhausen, Germany) at 10 W, 1.0 mbar, 60 s. The strips were then incubated in a solution of 1% v/v 2-[Methoxy(polyethyleneoxy)9-12propyl]trimethoxysilane in methanol for 1 h at room temperature (RT). After washing in methanol, the strips were vacuum dried for 1 h. Prior to cell culture, PI strips were washed in a solution of 70% ethanol, stretched over a CellCrown insert (inner open surface area: 0.58 cm^2^, Scaffdex Oy) and placed into a 24-well plate.

### 3.3. Hydrogel Synthesis

First, [2-(Methacryloyloxy)ethyl]dimethyl-(3-sulfopropyl) ammonium hydroxide (SBMA) was dissolved in deionized (DI) water at a molar concentration of 5 mol/L. The crosslinker, PEGDMA (0.1% w/w vs. monomer) and the initiator Irgacure I2959 (0.05% w/w by weight of the monomer) were added and completely dissolved at RT in dark conditions. Then, the solution was cast in a cylindrical silicon mold and photopolymerized for 20 min at 365 nm. After polymerization, the hydrogels were removed from the mold and immersed in a large volume of PBS. To ensure that nonreacted initiators or monomers were totally removed from the hydrogel, PBS was frequently replaced. Lastly, the poly(SBMA) hydrogels were left to swell in cell culture medium for 24 h [21].

### 3.4. Mechanical Characterization

Swollen hydrogels with cylindrical shapes (8 mm in diameter and 3 mm in thickness) were placed on the compression plate of an Instron tensile testing machine (model 3365, Instron, Norwood, MA, USA) equipped with a 10 N load. At least five cylindrical specimens were compressed up to 80% strain at a compressive strain rate of 1 mm/min. Young’s modulus was calculated from the quasi-linear portion of the first tract of the load curve.

### 3.5. Myofibroblast Differentiation and Immunofluorescence Analysis

Human dermal fibroblasts were cultured at 37 °C in a humidified atmosphere containing 5% CO_2_ in DMEM High Glucose, supplemented with 2 × 10^−3^ M L-glutamine, MEM nonessential amino acids solution, 100 IU/mL penicillin, 100 μg/mL streptomycin and 10% human serum, hereafter referred to as growth medium. For fibroblast differentiation into myofibroblasts, cells were stimulated with LPS (1 μg/mL) for 72 h in growth medium [82]. Cell differentiation was verified by immunofluorescence staining after fixation in 4% PFA in PBS and permeabilization in 0.1% Triton X-100, against: α-Smooth Muscle Actin (α-SMA, diluted 1:50 in BlockAid solution) incubated for 30 min at RT, Collagen Type I (Coll I, diluted 1:100 in BlockAid solution) incubated overnight (o.n.) at 4 °C [83,84,85]. Secondary antibodies were incubated for 45 min. at RT as follows: goat anti-mouse IgG Alexa Fluor-488 and goat anti-rabbit IgG Alexa Fluor-568, both diluted 1:400 in BlockAid solution. Nuclei were counterstained in blue with DAPI (1:10,000 dilution for 10 min. in PBS).

### 3.6. Cell Viability/Cytotoxicity Assay

Cell viability was assessed using Vybrant cytotoxicity assay as previously described [86]. Briefly, the release of the cytosolic enzyme glucose 6-phosphate dehydrogenase (G6PD) from damaged myofibroblasts into the surrounding medium was quantified after 24 h and 48 h in culture between the different experimental groups. Then 50 μL of supernatant from each specimen were transferred into a 96-well plate and, after 10 min incubation with 50 μL of resazurin/reaction mixture at 37 °C in 5% CO_2_, the fluorescent metabolite of resazurin (resorufin) was detected (ex. 530 nm; em. 570 nm) on a Tecan Infinite M200-Pro multiplate reader (Tecan, Männedorf, Switzerland).

For analysis of cell viability of myofibroblasts cultured on the hydrogel and polystyrene control surfaces after 1 and 4 days, a Live/Dead assay was done as in [87] according to manufacturer’s instructions (Thermo Fisher Scientific, Waltham, MA, USA), with living cells stained in green (FITC channel) and dead cells in red (Texas Red channel). For each channel, micrographs were acquired using a Nikon A1R+ laser scanning confocal microscope (Nikon, Tokyo, Japan) and processed by (semi) automated high-content image analysis tools (Nikon, Tokyo, Japan) for quantitative evaluation of live and dead cells in at least three independent wells for each sample and three to five random areas in each well. Results were plotted as the percentage of live cells vs. total cells for each experimental condition.

### 3.7. Cell Adhesion Assay

To measure cell adhesion and qualitatively evaluate cell morphology on the differ ent substrates, with and without the addition of human plasma fibrinogen used at a concentration of 4.5 mg/mL according to [39,40], myofibroblasts were stained in green either with CMFDA dye used on live cells according to manufacturer’s instructions, or with α-SMA antibody after fixation and permeabilization treatment. Nuclei were counterstained in blue with DAPI. For each channel, micrographs were acquired using a Nikon A1R+ laser scanning confocal microscope (Nikon, Tokyo, Japan) and processed by (semi) automated high-content image analysis tools (Nikon, Tokyo, Japan). To quantitatively evaluate cell adhesion, at least three independent wells were analyzed for each sample by acquiring three to five random areas in each well and comparing them to internal controls. To show the growing properties of myofibroblasts on the hydrogel surface, hydrogels with stained cells (after 24 h in culture) were cut along a sagittal plane, overturned and maintained in culture medium to capture micrographs under confocal microscope at 20× magnification.

### 3.8. THP-1 Cell Culture and Macrophage Differentiation

The human monocytic cell line THP-1 was grown in suspension into T-75 cm^2^ cell culture flasks (Corning Inc., New York, NY, USA) at subculture concentrations between 2–5 × 10^5^ viable cells/mL in complete growth RPMI-1640 medium supplemented with 2 × 10^−3^ M L-glutamine, MEM nonessential amino acids solution, 100 IU/mL penicillin, 100 μg/mL streptomycin and 10% fetal bovine serum (FBS) at 37 °C in a humidified atmosphere containing 5% CO_2_. Suspended cells were split by centrifugation and the cell pellet was resuspended in fresh growth medium. To induce their differentiation into nonpolarized M_0_ macrophages (hereafter referred to as non-stimulated cells), cells were cultured in adhesion through a stimulation with 200 nm PMA in complete growth medium for 48 h [88]. Then, the PMA-differentiated and adherent M_0_ macrophages were detached by 0.25% trypsin/0.02% EDTA solution (Merck) and seeded on hydrogel and ctrl polystyrene surfaces at a density of 2.5 × 10^4^/cm^2^ in complete DMEM High Glucose supplemented with 10% human serum for 24 h in culture to allow adhesion. The next day, growth medium was replaced with fresh complete DMEM High Glucose supplemented with 10% human serum for M_0_ polarization to M_1_ macrophages (hereafter referred to as stimulated cells) by means of 100 ng/mL LPS + 20 ng/mL IFN-γ for 24 h [54,57].These were then used for flow cytometry analysis and oxidative stress experiments. Control M_0_ macrophages were grown in complete medium without stimulation with LPS and IFN-γ.

### 3.9. Flow Cytometry

M_0_ and stimulated macrophages were stained with the PE-conjugated monoclonal mouse anti-human CD68 antibody and the FITC-conjugated monoclonal mouse anti-human CD86 antibody (ImmunoTools, Friesoythe, Germany) according to manufacturer’s instructions. After specific stimulations (as reported above in Section 3.8) for M_0_ and M_1_ differentiation, 2 × 10^5^ cells/sample were harvested and resuspended in PBS without Ca^2+^ and Mg^2+^. Dead cells were labeled using the fixable viability dye eFluor 780 (Thermo Fisher Scientific, Waltham, MA, USA) in PBS without Ca^2+^ and Mg^2+^ for 30 min in ice, and cell viability analysis was done according to manufacturer’s instructions. Nonspecific antigens were blocked by 2% FBS in PBS and samples stained for 30 min in ice with the above antibodies. For intracellular CD68, fixation in 4% PFA in PBS and permeabilization in 0.1% Triton X-100 were done prior to staining. After staining, cells were washed in PBS without Ca^2+^ and Mg^2+^ and analysis was performed using a CytoFLEX flow cytometer (Beckman Coulter, Brea, CA, USA). Analysis of endogenous fluorescence in each cell type was used as a negative control. Data on CD86 expression levels were reported as the ratio between M_1_ and M_0,_ each normalized against its own negative control.

### 3.10. Analysis of Oxidative Stress

To investigate and compare oxidative stress levels in stimulated pro-inflammatory macrophages and non-stimulated M_0_ macrophages between zwitterionic and control polystyrene surfaces the Total ROS/RNS Detection kit (ENZO Life Sciences, Farmingdale, New York, NY, USA) was used according to manufacturer’s instructions. Briefly, after specific stimulations, live cells were incubated with the ROS/RNS (green fluorescence) detection reagent at a final concentration of 3 μM in the wash buffer provided with the kit for 1 h in the dark at 37 °C. After incubation, cells were washed in wash buffer and immediately observed. Pyocyanin (i.e., ROS/RNS inducer) incubated at 500 μM was used as a positive ctrl [89]. Quantitative analysis was performed by (semi) automated high-content image tools on micrographs captured with a confocal Nikon A1R+ microscope (Nikon, Tokyo, Japan) by taking at least 3 random microscopic fields per well and applying the same regions of interest (ROIs) to all samples. For cell imaging, all nuclei were counterstained with the blue-fluorescent nuclear dye Hoechst 33342 (5 μg/mL in PBS). Fluorescence data in the graph were plotted as mean fluorescence intensity (MFI) levels of the ROIs occupied by the cells in each sample and reported as mean ± standard error of the mean (SEM) of three independent replicates.

### 3.11. Drug Release Profile

For Everolimus release from hydrogels, an overall 50.0 μg of drug according to [73], previously resuspended following manufacturer’s instructions (i.e., 30% Propylene glycol + 5% Tween 80 + ddH_2_0, hereafter referred to as resuspension solution), were loaded into the hydrogel in the dry state through surface adsorption. Then, exploiting the hydrogel swelling behavior, the gradual release of the drug was measured through absorbance readings at 280 nm, using an Infinite M200-Pro multiplate reader, TECAN (Männedorf, Switzerland). The Everolimus-loaded hydrogels were placed in a multiwell tissue culture plate and allowed to exchange against PBS (2.0 mL final volume so as to have 25.0 μg/mL of Everolimus) at 37 °C in a humidified atmosphere containing 5% CO_2_. At defined time points, aliquots (3 × 100 μL) were withdrawn and pipetted into 96-well plates for absorbance readings. At each withdrawal, half of the volume of the elution buffer was replenished with fresh PBS to avoid mass transfer equilibrium between the hydrogels and the surrounding environment. Cumulative release from hydrogels was normalized against internal controls (i.e., hydrogels loaded with equal volumes of the resuspension solution as specified above for the drug) and determined according to the Everolimus standard calibration curve. Data were reported as mean ± standard deviation (SD) of three independent replicates.

### 3.12. Hydrogel Coating of the Polyimide (Kapton) Surface

Kapton stripes were treated with oxygen plasma for 5 min in a FEMTO plasma cleaner. Then, the zwitterionic hydrogel solution, prepared as previously described, was poured onto the treated surface and exposed to UV light (365 nm) for 20 min. After irradiation, samples were washed in distilled water and dried o.n. at RT and atmospheric pressure prior to characterization. Control samples without plasma treatment were also analyzed for comparison.

### 3.13. Water Contact Angle

The water contact angles on PEG-coated PI, thin hydrogel film and nude PI (Kapton) surfaces were measured using the sessile drop technique with a CCD camera and processed by the Image J water contact angle plugin. Dried PEG coating, hydrogel and Polyimide surfaces were wetted with a 3 μL water drop. For each sample, 3 independent measurements were performed.

### 3.14. Software for Scientific Graphics

For illustrations and cartoons in Figure 1, Figure 2, Figure 5 and Figure 6 as previously detailed in figure captions, the BioRender.com software (accessed on 1 April 2022) was used according to the software citation policy and publishing license.

### 3.15. Statistical Analysis

Data were analyzed using GraphPad Prism ver. 8.4.3 software and reported as mean ± SD, unless otherwise specified, of at least three independent experiments. Student’s *t*-test and one-way analysis of variance (ANOVA), followed by Tukey’s multiple comparisons test, were used to assess statistical significance, which was set at 0.05.

## 4. Conclusions

Overall, our poly(SBMA) zwitterionic hydrogel showed lower cell adhesion and activation as well as different cell morphologies compared to adherent controls on polystyrene surfaces. It also exhibited the following advantageous features: (i) low stiffness, typical of soft hydrogels, with a Young’s modulus that reflected that of neural tissue; (ii) very high biocompatibility; (iii) ultra-low fouling characteristics, at least as good as PEG coatings, which enabled very low adhesion and activation of pro-fibrotic and pro-inflammatory cells; (iv) high hydrophilicity, showing reduced water contact angle (~30°) compared to pristine hydrophobic PI surfaces; (v) lower macrophage activation, reflected in reduced oxidative stress levels; (vi) sustained release of anti-fibrotic drugs in the context of FBR.

These hydrogel properties and known in vitro behaviors (i.e., gel swelling ability, anti-inflammatory characteristics and high intrinsic ionic conductivity due to the zwitterionic counterions) could be harnessed in vivo to reduce the fibrotic response associated with FBR. This poly(SBMA) zwitterionic hydrogel could also facilitate a gradual and controlled drug release around neural implants in future works. Our findings confirmed that, as alternatives to traditional PEG or PEGylated surfaces, zwitterionic polymers held great potential as suitable dressings of intraneural electrodes for in vivo strategies aimed at slowing down or halting FBR. Among other advantages, they are less susceptible to oxidative damage than PEG coatings in long-term neural implants.

## Figures and Tables

**Figure 1 molecules-27-03126-f001:**
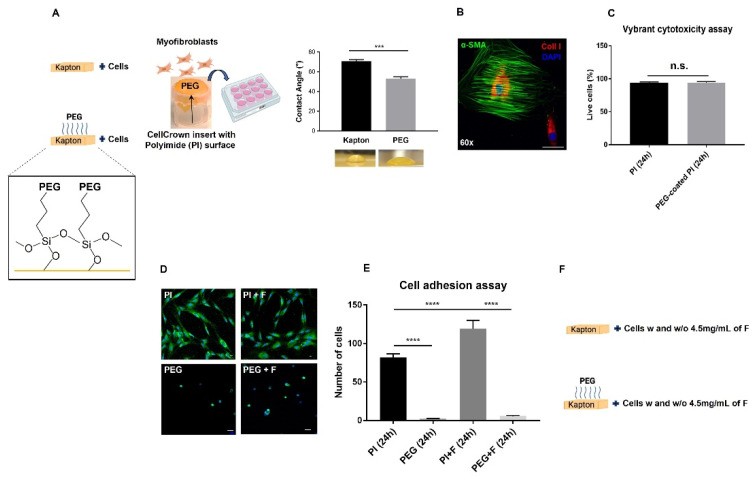
**Biocompatibility of PEG coating and cell adhesion assay between PI and PEG-coated PI surfaces.** (**A**) Experimental scheme of the functionalization of the Kapton surface with PEG coating (depiction of the chemistry is magnified in the inset and measurement of the water contact angle with representative photos is shown on the far right; data are plotted as mean ± SD of three independent experiments, *** *p* < 0.001) and cell seeding. (**B**) Representative immunofluorescence confocal image of activated myofibroblast (using 1 μg/mL of LPS for 72 h) stained in green for alpha- smooth muscle actin (α-SMA) and in red for collagen type I (Coll I). Nuclei are counterstained with DAPI; magnification 60×, scale bar = 50 μm. Cell viability/cytotoxicity analysis (**C**), using the Vybrant cytotoxicity assay, on activated myofibroblasts after 24 h in culture showing very high and similar cell viability levels between nude Kapton (PI) surface ((**A**), top cartoon) and PEG-coated PI surface (**A**, bottom cartoon) stretched over the CellCrown inserts placed into a tissue culture multiwell plate ((**A**), middle cartoon). (**D**) Representative confocal images of activated myofibroblast (stained in green with α-SMA and nuclei counterstained in blue with DAPI). Different cell morphology is observed on PEG-coated surfaces vs. PI surfaces due to high PEG hydrophilicity, in agreement with data from the literature; magnification 20×, scale bar = 20 μm. Cell adhesion assay (**E**), measured by (semi) automated high-content image analysis, on activated myofibroblasts after 24 h in culture on nude PI (Kapton) and PEG-coated PI surfaces (**F**) stretched over the CellCrown inserts as in (**A**), with and without human plasma fibrinogen (**F**) at 4.5 mg/mL (as shown in representative confocal images in (**D**)). Except (**A**), all data are plotted as mean ± SEM of three independent experiments, **** *p* < 0.0001. (**A**,**F**) created with BioRender.com (accessed on 1 April 2022).

**Figure 2 molecules-27-03126-f002:**
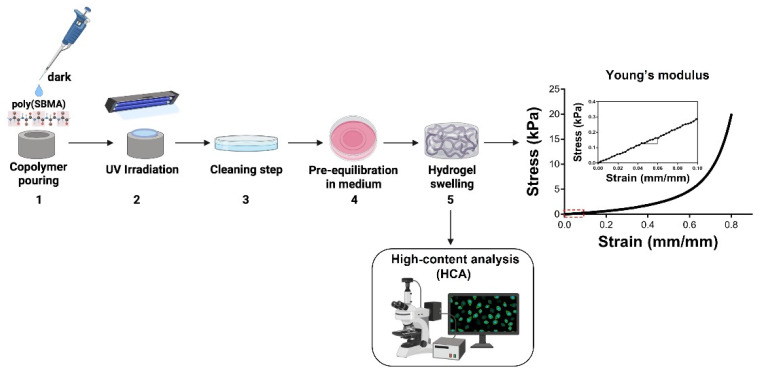
**Synthesis, mechanical characterization and high-content analysis of the poly(SBMA) zwitterionic hydrogel.** (1) The solution is cast in a cylindrical silicon mold and (2) photopolymerized for 20 min. at 365 nm. (3) Cylindrical hydrogel is removed from the mold, washed in PBS and (4) pre-equilibrated in cell culture medium for 24 h; (5) Swollen hydrogel is then used for biological (HCA) and mechanical characterization. The Young’s modulus (i.e., 2.7 ± 0.24 kPa, shown in the inset of the far-right stress-strain curve as a magnification of the red-dashed rectangular area) reflects that of the neural tissue. Created with BioRender.com (accessed on 1 April 2022) except the Young’s modulus curve.

**Figure 3 molecules-27-03126-f003:**
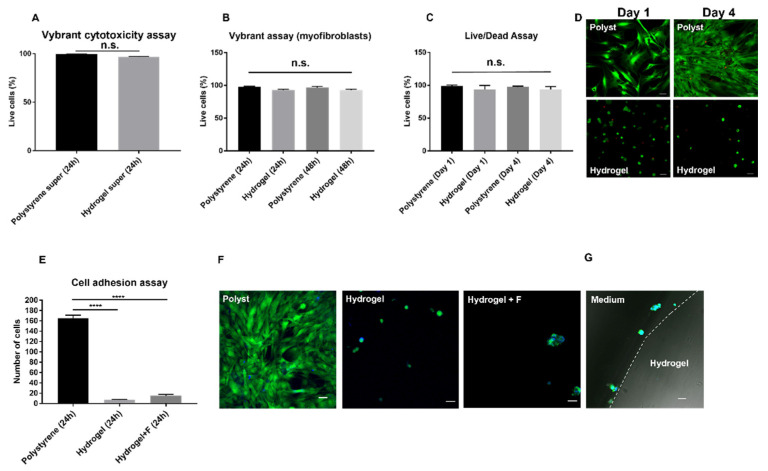
**Biocompatibility tests of the poly(SBMA) zwitterionic hydrogel, Live/Dead assay****and cell adhesion assay between polystyrene and hydrogel****surfaces.** Cell viability/cytotoxicity analysis, using the Vybrant cytotoxicity assay, studying the effects of polystyrene (ctrl)- and hydrogel-derived supernatants (super) on activated myofibroblasts after 24 h in culture (**A**). Myofibroblast viability after 24 h and 48 h in culture on polystyrene (ctrl) tissue culture plates and hydrogel surfaces (**B**). Live/Dead assay (**C**) on myofibroblasts cultured on polystyrene (ctrl) tissue culture plates and hydrogel surfaces for 1 and 4 days with representative confocal images in (**D**), showing live cells in green and dead cells in red and having diverse morphology and growth characteristics between the two surfaces; magnification 20×, scale bar = 50 μm. Data are plotted as mean ± SEM of three independent experiments. Cell adhesion assay (**E**), measured by (semi) automated high-content image analysis, on activated myofibroblasts after 24 h in culture between polystyrene (ctrl, with a high number of cells spread over the whole surface) and poly(SBMA) zwitterionic hydrogel surfaces with and without human plasma fibrinogen (**F**) at 4.5 mg/mL (showing very low adherence and round morphology characteristic of non-activated cells, with a few isolated clusters in sporadic areas of the gel), as shown in representative confocal images in (**F**) (myofibroblasts are stained in green with the CMFDA dye and nuclei counterstained in blue with DAPI); magnification 20×, scale bar = 50 μm. Data are plotted as mean ± SEM of three independent experiments, **** *p* < 0.0001. Representative micrograph in (**G**) shows the interface between culture medium (left) and hydrogel surface (right). Note myofibroblasts growing aligned on the hydrogel surface, but not in the thickness of the gel, with no penetration (cells were stained as described above); magnification 20×, scale bar = 20 μm.

**Figure 4 molecules-27-03126-f004:**
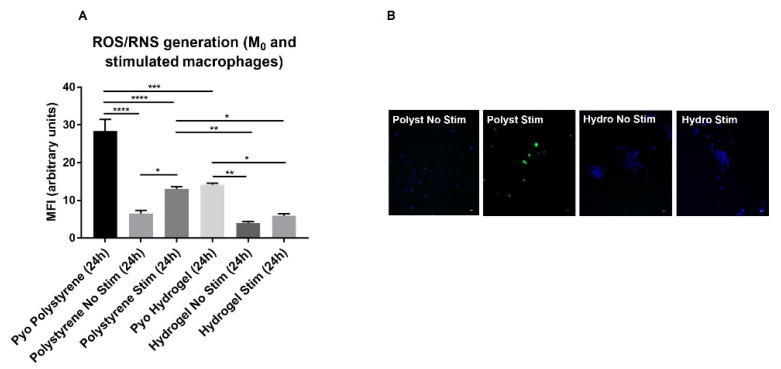
**Activation of human M_0_ macrophages and stimulated macrophages evaluated through the measurement of oxidative stress production between polystyrene and hydrogel surfaces.** (**A**) HCA of the intracellular levels of ROS/RNS (reactive oxygen/nitrogen species) in macrophages after 24 h in culture; Pyo: Pyocyanin (ROS/RNS inducer was used as positive ctrl); No stim: M_0_ macrophages with no stimulation; Stim: macrophages stimulated with LPS + IFN-γ; (**B**) Representative confocal images of M_0_ and stimulated macrophages, cultured on polystyrene and hydrogel surfaces, stained in green through the ROS/RNS detection reagent; nuclei are counterstained in blue with Hoechst dye; magnification 20×, scale bar = 20 μm. Data are plotted as mean ± SEM of three independent experiments and expressed as MFI: mean fluorescence intensity, * *p* < 0.05, ** *p* < 0.01, *** *p* < 0.001, **** *p* < 0.0001.

**Figure 5 molecules-27-03126-f005:**
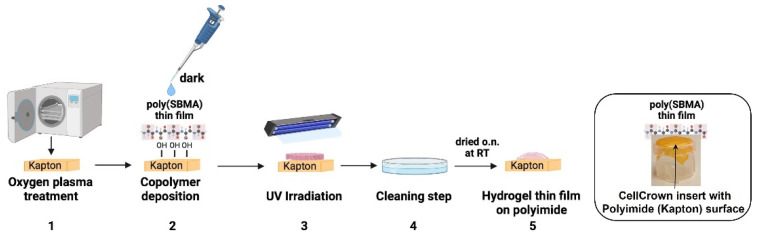
**Protocol for coating the Polyimide surface with a thin film of the zwitterionic hydrogel.** (1) Kapton stripes are treated with oxygen plasma. (2) A copolymer solution is deposited onto the Kapton surface, followed by (3) UV irradiation for 20 min and (4) extensive washing in distilled water. Finally, it is dried overnight at RT to obtain a coating made of a hydrogel thin film (5), which is eventually stretched over CellCrown inserts for cell culture experiments. Created with BioRender.com (accessed on 1 April 2022).

**Figure 6 molecules-27-03126-f006:**
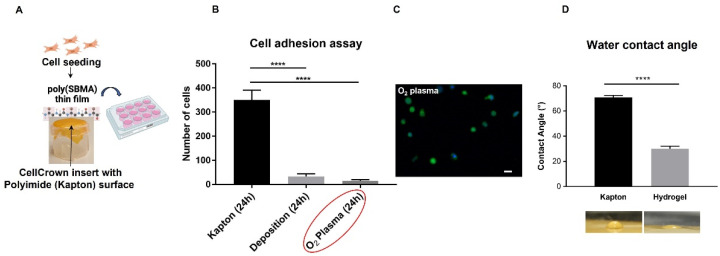
**Cell adhesion assay on a thin film of the poly(SBMA) zwitterionic hydrogel and analysis of the water contact angle.** (**A**) Seeding of activated myofibroblasts on a thin film of the poly(SBMA) hydrogel-coated PI (Kapton) surface, stretched over the CellCrown insert and placed into a tissue culture multiwell plate. Cell adhesion assay (**B**), measured by (semi) automated high-content image analysis, on activated myofibroblasts after 24 h in culture on a nude hydrophobic Kapton surface compared with thin films of the hydrogel obtained using different methods: direct deposition vs. O_2_ plasma plus photochemical treatment (circled in red), which showed the lowest number of adherent cells, with round shapes, as observed in the representative confocal image in (**C**). Myofibroblasts are stained in green with the CMFDA dye and nuclei are counterstained in blue with DAPI; magnification 20×, scale bar = 20μm. Measurement of the water contact angle (**D**) with a layer of the hydrophilic zwitterionic hydrogel, grafted through the O_2_ plasma surface activation (bottom right photo), and without (bottom left photo) coating of the hydrophobic Kapton surface (from 70° on nude Kapton to 30° on the hydrogel layer). Data are plotted as mean ± SD of three independent experiments, **** *p* < 0.0001. (**A**) Created with BioRender.com (accessed on 1 April 2022).

**Figure 7 molecules-27-03126-f007:**
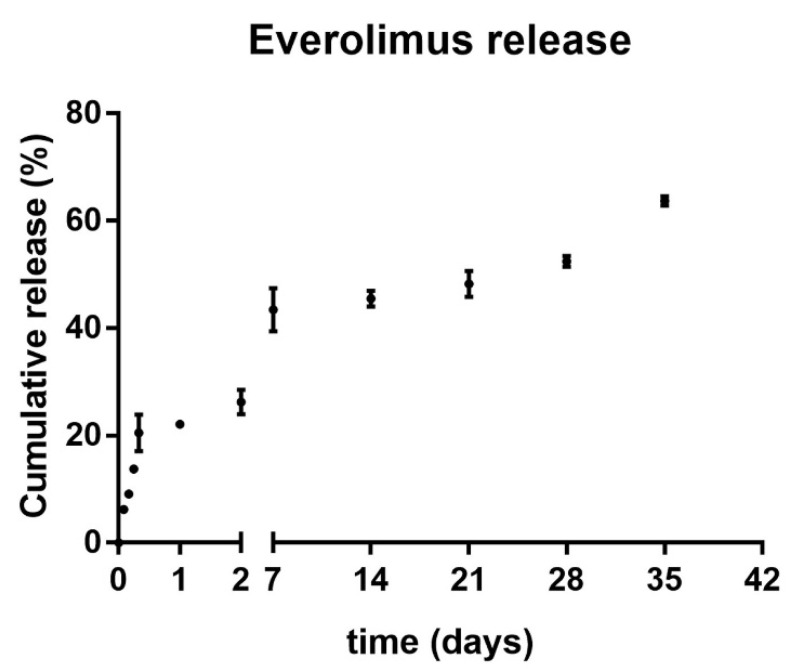
**In vitro release profile of Everolimus from the zwitterionic hydrogel.** The slope of the cumulative release of the Everolimus is represented as a percentage of the total drug payload vs. time expressed in days (with x-axis break); experiments performed in triplicate and plotted as mean ± SD.

## Data Availability

Data are contained within the article and Appendix A.

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
