# Peer review of "A Soft Zwitterionic Hydrogel as Potential Coating on a Polyimide Surface to Reduce Foreign Body Reaction to Intraneural Electrodes"

_molecules, 2022, doi:10.3390/molecules27103126_

Round 1
Reviewer 1 Report
Gori et al. present the coating of a polyimide surface with poly(sulfobetaine methacrylate) soft zwitterionic hydrogels and study cell adhesion and biocompatibility on the engineered surfaces. The use of the hydrogel for the delivery of the immunomodulatory molecule Everolimus is also studied. The obtained soft zwitterionic hydrogels are appealing materials for preventing foreign body reaction to intraneural electrodes and could be an improvement in comparison to the standard poly(ethylene glycol) coatings. The manuscript will be of interest for the audience of Molecules.
The work is well presented, organized and clear. The materials and methods section is detailed. I recommend the article for publication after minor revision.
Minor revision and suggestions:
-Minor English corrections are necessary (e.g., excess of commas).
-Avoid using phrases as ‘have never been’ (line 93) or ‘for the first time’ (line 101). Try to rewrite them to something similar to ‘To the best of our knowledge’ as you did in line 336.
-The first time that Figure 1 is mentioned is in line 112 and it is referring to Figure 1C. Is it possible to reorganize the Figure so it follows the mention order in the main text?
-Please, indicate which portions of the figure were created with BioRender.com (C and F for Figure 1?).
-Is it possible to explain how you calculated the % of live cells from the Vybrant cytotoxicity assay? Is this assay also considering non-adhered cells?
-Can you explain why you used 4.5 mg/mL of Fibrinogen? I see that is related to references 35 and 36, but I would appreciate more context.
-In line 153-154 you have [] instead of ().
-In lines 160-162 you mention ‘with a few round cells on PEG-coated PI, which is the typical shape of non-activated cells, compared to those on pristine PI that were well spread over the whole surface…’. What does non-activated mean here? Is rounded the typical shape of non-activated cells or is just the expected shape for cells with poor adhesion on the PEG-coated surface?
-In lines 213-215 it says, ‘Taken together, these results clearly demonstrate a low capacity of adhesion and survival of myofibroblasts in long-term cultures on the zwitterionic polymer…”. As it is written, it appears to say that the myofibroblasts have low capacity of survival, but a ~100% viability is shown in Figure 3. Maybe the sentence can be improved.
-In lines 246-247 it says, ‘shows the interface between culture medium (left) and hydrogel surface (right)’, but the right is showing the inner or bulk of the hydrogel, right?
-Figure 3 shows no cytotoxicity of the hydrogel. Does the total cell number change from day 1 to day 4? If it does, does it change in the same amount for the polystyrene and the hydrogel?
-In lines 260-262 it says, ‘non-biofouling properties due to the high surface hydration of zwitterionic chemistries that hinders hydrophobic interactions with cell membranes’. And what about cell adhesive proteins such as fibronectin or vitronectin that mediates surface-cell interactions?
-In lines 302-303 it says, ‘Stim: pro-inflammatory M1 macrophages stimulated with LPS + IFN-γ’. Was the subpopulation of M1 macrophages sorted and used for the experiments or it was just the complete population of macrophages that was stimulated with LPS + IFN-γ? If it is the second, maybe ‘Stim: macrophages stimulated with LPS + IFN-γ’ is more correct.
-In lines 387-399 it says, ‘that showed the lowest number of adherent cells’. Is this statistically significant?
-I would mention in materials and methods sections when the staining and imaging was performed with live cells or with fixed cells. For example, for section 3.5 I assume that the cells were fixed and permeabilized, but for the experiments with CMFDA?
Author Response
Comments and Suggestions for Authors:
Gori et al. present the coating of a polyimide surface with poly(sulfobetaine methacrylate) soft zwitterionic hydrogels and study cell adhesion and biocompatibility on the engineered surfaces. The use of the hydrogel for the delivery of the immunomodulatory molecule Everolimus is also studied. The obtained soft zwitterionic hydrogels are appealing materials for preventing foreign body reaction to intraneural electrodes and could be an improvement in comparison to the standard poly(ethylene glycol) coatings. The manuscript will be of interest for the audience of Molecules.
The work is well presented, organized and clear. The materials and methods section is detailed. I recommend the article for publication after minor revision.
Minor revision and suggestions:
- We sincerely acknowledge the Reviewer for the interesting and useful suggestions that improve the quality of the manuscript and we made corrections accordingly, with the hope to have exhaustively addressed the Reviewer’s comments.
-Minor English corrections are necessary (e.g., excess of commas).
- We made corrections to different sections of the manuscript including commas.
-Avoid using phrases as ‘have never been’ (line 93) or ‘for the first time’ (line 101). Try to rewrite them to something similar to ‘To the best of our knowledge’ as you did in line 336.
- Corrections to the text have been made as requested in lines 115-120.
-The first time that Figure 1 is mentioned is in line 112 and it is referring to Figure 1C. Is it possible to reorganize the Figure so it follows the mention order in the main text?
- Figure 1 has been reorganized according to the main text and figure caption, as requested. Furthermore, we added new data in Figure 1 according to the requests of Reviewers n. 2 and n. 3.
-Please, indicate which portions of the figure were created with BioRender.com (C and F for Figure 1?).
- All portions of the Figures 1, 2, 5 and 6, created with BioRender.com, have been specified in the captions.
-Is it possible to explain how you calculated the % of live cells from the Vybrant cytotoxicity assay? Is this assay also considering non-adhered cells?
- This assay was carried out to confirm the biocompatibility of the PEG coating, similarly to the PI [27-29], according to the literature [30-32]. The Vybrant assay evaluates the total amount of G6PD released in the cell culture supernatant from damaged cells. Since many cells, due to the low adherence provided by the PEG surface, detach from the coating, after waiting a few hours post seeding to allow cell adhesion on the PEG surface, we changed medium with fresh one so as to evaluate the total G6PD released only by adherent cells after 24h in culture. Thus, that’s why we observed very high level of cell viability that is comparable with the PI surface in Figure 1C in which data were represented as % of live cells = 100% viability - % of dead cells.
-Can you explain why you used 4.5 mg/mL of Fibrinogen? I see that is related to references 35 and 36, but I would appreciate more context.
- The addition of human plasma fibrinogen was done for mimicking more closely an in vivo physiological condition of the plasma proteins that are present in human blood and that herein may affect cell adhesion to the surface, as also stated in a new sentence in lines 200-201; we used this concentration which is the highest concentration circulating in human plasma (i.e., 4.0-4.5 mg/mL) as described in the literature [Tennent G.A. et al. Blood 2007, 109(5):1971-4, doi: 10.1182/blood-2006-08-040956; Schlimp C.J. et al. J Trauma Acute Care Surg. 2015, 78(4):830-36, doi: 10.1097/TA.0000000000000546]. For the sake of clarity, these new references [37,38] have been added to the manuscript.
-In line 153-154 you have [] instead of ().
- In lines 196-197 of the new version of the manuscript, the [ ] were used as there is already (α-SMA) in the same sentence.
-In lines 160-162 you mention ‘with a few round cells on PEG-coated PI, which is the typical shape of non-activated cells, compared to those on pristine PI that were well spread over the whole surface…’. What does non-activated mean here? Is rounded the typical shape of non-activated cells or is just the expected shape for cells with poor adhesion on the PEG-coated surface?
- It is known from the literature that different cell types, such as fibroblasts and macrophages, show not only a rounded shape due to minimal adhesion on PEG-based hydrogels but this in turn results also in a reduced metabolic activation/response and low expression levels of genes involved in the production of inflammatory cytokines, giving rise to less severe FBR in vivo and in vitro [11-12,23]. Thus, a rounded shape for such cell type (i.e., fibroblasts) that normally grows well spread and adherent, also means low- or no-activation. We added a new sentence to explain this further in line 205 of the revised version of the manuscript.
-In lines 213-215 it says, ‘Taken together, these results clearly demonstrate a low capacity of adhesion and survival of myofibroblasts in long-term cultures on the zwitterionic polymer…”. As it is written, it appears to say that the myofibroblasts have low capacity of survival, but a ~100% viability is shown in Figure 3. Maybe the sentence can be improved.
- The sentence, now in lines 259-260, has been modified according to the Reviewer’s suggestion.
-In lines 246-247 it says, ‘shows the interface between culture medium (left) and hydrogel surface (right)’, but the right is showing the inner or bulk of the hydrogel, right?
- The hydrogel was cut along a sagittal plane and overturned to show the inner section of the gel (i.e., from top surface on the left, with dashed line, to the base on the right) with the aim to demonstrate no myofibroblast penetration and growth into the thickness of the gel as explained in section 3.7 of Materials & Methods as well as in the main text at page 9 in lines 294-302 of the new version of the manuscript.
-Figure 3 shows no cytotoxicity of the hydrogel. Does the total cell number change from day 1 to day 4? If it does, does it change in the same amount for the polystyrene and the hydrogel?
- Taking into account the total number of adherent cells on the two surfaces, the percentage of live cells shown in the graph remains basically the same between day 1 and day 4 both for the hydrogel and polystyrene, with no statistically significant differences. As expected, the total number of cells on the polystyrene increases between day 1 and day 4 due to normal cell proliferation on the control surface, whereas it decreases on the hydrogel due to the lower adhesion on its hydrophilic and soft surface.
-In lines 260-262 it says, ‘non-biofouling properties due to the high surface hydration of zwitterionic chemistries that hinders hydrophobic interactions with cell membranes’. And what about cell adhesive proteins such as fibronectin or vitronectin that mediates surface-cell interactions?
- Regarding cell adhesive proteins, we referred to data from the literature: it has been previously demonstrated [21,47] that the SBMA zwitterionic chemistry adsorbs a very low amount of proteins from 100% blood plasma or serum [47]. Furthermore, by means of an ELISA assay, the authors also showed excellent antifouling properties of the same hydrogel with low protein adsorption (< 10%) compared to tissue culture polystyrene [21]. Taking these data into account, we decided not to repeat the same experiments. Reference 47 has now been added to the manuscript and changes to the text made accordingly in lines 310-311.
-In lines 302-303 it says, ‘Stim: pro-inflammatory M1 macrophages stimulated with LPS + IFN-γ’. Was the subpopulation of M1 macrophages sorted and used for the experiments or it was just the complete population of macrophages that was stimulated with LPS + IFN-γ? If it is the second, maybe ‘Stim: macrophages stimulated with LPS + IFN-γ’ is more correct.
- The complete population of macrophages was used after stimulation, thereby we changed the main text, Figure 4 and caption according to the Reviewer’s suggestion although stimulated pro-inflammatory M1 macrophages are commonly referred in the literature to as polarized cells toward the M1 state using the herein classical differentiation protocol (i.e., in response to stimulation with microbial factors, such as LPS and IFN-γ) [54, Mantovani A, Sica A, Sozzani S, et al. Trends Immunol. 2004;25(12):677–86; Martinez FO, Gordon S, Locati M, Mantovani A. J Immunol. 2006;177(10):7303–11; Rey-Giraud F, Hafner M, Ries CH. PLoS One. 2012;7(8), e42656] and confirmed only by flow cytometry [54-56].
-In lines 387-399 it says, ‘that showed the lowest number of adherent cells’. Is this statistically significant?
- Although the difference is not statistically significant (mean values: 34±10.4 for deposition vs. 15±5.3 for O2 plasma), the choice of the O2 plasma treatment was due to the higher stability, in long-term cultures, of the polymer coating obtained with the latter method, compared with the physical adsorption obtained through the deposition method.
-I would mention in materials and methods sections when the staining and imaging was performed with live cells or with fixed cells. For example, for section 3.5 I assume that the cells were fixed and permeabilized, but for the experiments with CMFDA?
- We apologize for the missing information; we made corrections according to Reviewer’s suggestion in sections 3.5, 3.7 and 3.10.
Reviewer 2 Report
The authors reported a study where a comparison between synthesized organic zwitterionic coating based on poly(sulfobetaine methacrylate) [poly(SBMA)] hydrogel and the synthetic coating based on PEG. The materials are planned for the reduction of adhesion and activation of inflammatory and fibrogenic cells, early hallmarks of the FBR, to Polyimide surfaces. The manuscript is well supported by data and organized in the manner to prove the expected outcome. However, there are still unclear things which require reformulation and details to be inserted.
Please find below the following comment/suggestions with may contribute to the quality improvement of the manuscript.
- The authors provided an abstract which looked very general formulated and provided very few details from the study. Briefly, the authors should focus on purpose of the study, materials used, methods developed, and outcome obtained. The authors are advised to reformulate so that the abstract part to be attractive and concise.
- The authors provided somehow the purpose of the study, but it is not enough highlighted; the authors should reformulate the part where they state the innovative aspect of the paper and their aim.
- Page 5 line 194, the phrase does not look finished or there is no subject; the authors should clarify or reformulate.
- Identically page 6 line 251, the authors should double check the phrase.
- The authors have used a software Biorender but the origin and what is it for was not clearly explained; the authors are advised to explain in a more concise way the role of this software.
- The authors stated “Poly(dimethylsiloxane-b-ethylene oxide) methyl terminated was purchased from Polysciences Inc”; however, details on the polymers were not provided such as molecular weight, etc.
- İn the case of “Functionalization of Polyimide (Kapton) surface with PEG.”; the authors mentioned the synthesis of the materials, but no evidence was provided (structural or other); yield of reaction should be mentioned as well.
- In the case of hydrogel, an external crosslinking agent was used; in this case a crosslinking degree should be determined.
Author Response
The authors reported a study where a comparison between synthesized organic zwitterionic coating based on poly(sulfobetaine methacrylate) [poly(SBMA)] hydrogel and the synthetic coating based on PEG. The materials are planned for the reduction of adhesion and activation of inflammatory and fibrogenic cells, early hallmarks of the FBR, to Polyimide surfaces. The manuscript is well supported by data and organized in the manner to prove the expected outcome. However, there are still unclear things which require reformulation and details to be inserted.
Please find below the following comment/suggestions with may contribute to the quality improvement of the manuscript.
- We sincerely acknowledge the Reviewer for the interesting and useful suggestions that improve the quality of the manuscript and we made corrections accordingly, with the hope to have exhaustively addressed the Reviewer’s comments.
1. The authors provided an abstract which looked very general formulated and provided very few details from the study. Briefly, the authors should focus on purpose of the study, materials used, methods developed, and outcome obtained. The authors are advised to reformulate so that the abstract part to be attractive and concise.
- We thank the Reviewer for this suggestion and tried to improve the quality of the abstract, which is now more straightforward and also shorter.
2. The authors provided somehow the purpose of the study, but it is not enough highlighted; the authors should reformulate the part where they state the innovative aspect of the paper and their aim.
- Following the Reviewer’s suggestion, we also changed the last section of the Introduction (from line 121 to 135 of the new version of the manuscript) and lines 710-711 in the Conclusions to stress a little more the main purpose and outcomes of the study as well as the innovation. We acknowledge the Reviewer for this useful suggestion that now we think it may have improved the final message and the aim of our study.
3. Page 5 line 194, the phrase does not look finished or there is no subject; the authors should clarify or reformulate.
- The sentence is interrupted by the insertion of Figure 2 within the main text, close to its first citation, as requested by the Author guidelines of the Journal. Therefore, the subject (i.e., supernatant obtained from the polystyrene surface) is not missing but it is now in line 240 of the revised version of the manuscript.
4. Identically page 6 line 251, the authors should double check the phrase.
- Same as for line 240, the subject (i.e., representative micrograph) is now at page 8 in line 271 of the revised version of the manuscript, as the sentence is interrupted by Figure 3.
5. The authors have used a software Biorender but the origin and what is it for was not clearly explained; the authors are advised to explain in a more concise way the role of this software.
- We apologize for the missing information about the software and according to the Reviewer’s suggestion we introduced the new section 3.14 Software for scientific graphics in Materials and Methods of the new version of the manuscript.
6. The authors stated “Poly(dimethylsiloxane-b-ethylene oxide) methyl terminated was purchased from Polysciences Inc”; however, details on the polymers were not provided such as molecular weight, etc.
- We inserted the MW and density of the correct polymer, that is 2-[Methoxy(polyethyleneoxy)9-12propyl]trimethoxysilane in the section Materials and Methods (3.1 Chemicals and cells) of the new version of the manuscript in lines 491-492. Properties of the above polymer are as follows:
MOLECULAR WEIGHT: 591-719 g/mol
RELATIVE DENSITY: 1.071 g/mL
For all chemical reagents, the Mn and density (when available in liquid form) have been inserted into the manuscript in lines 503-506 under the section Materials and Methods, according to the Reviewer’s suggestion.
7. İn the case of “Functionalization of Polyimide (Kapton) surface with PEG.”; the authors mentioned the synthesis of the materials, but no evidence was provided (structural or other); yield of reaction should be mentioned as well.
- As far as the functionalization with PEG is concerned, the mechanism was as follows: following hydrolysis of methoxy silane groups, PEG-silane is grafted to plasma-treated Kapton surface via condensation of formed silanols with surface hydroxyl groups in methanol. Further stabilization occurs via siloxane bonds between immobilized PEG-silane chains. This description has been inserted in the new version of the manuscript at page 4 (lines 144-147) together with a sketch of the chemistry in the new Figure 1A and caption. Furthermore, we also inserted the water contact angle of the PEG-coated PI surface in the main text at page 4 (lines 148-151) with corresponding images and graph in Figure 1A.
8. In the case of hydrogel, an external crosslinking agent was used; in this case a crosslinking degree should be determined.
- As claimed in our manuscript, we used the same crosslinking agent (i.e., PEGDMA) and same synthesis process of [21] in which a characterization of the polySBMA zwitterionic hydrogel was already provided although for a different application (i.e., skin regeneration). Moreover, different crosslinking degrees were tested by the same authors [41] for evaluating the effect on the physicochemical properties of the hydrogel in wound healing applications. Therefore, we referred to the literature as such analysis was not within the scope of the present work.
Reviewer 3 Report
The research article by Gori et al. is well presented. All experiments necessary for indicating the effect of developing soft zwitterionic hydrogel coatings for reducing the foreign body reaction is well conducted and nicely presented through figures. However, there are some minor changes is to be done in order to improve the manuscript.
- Typos. Line 78-79, 'Poly(ethylene glycol) (PEG)' to 'poly(ethylene glycol) (PEG)'; line 410, 'TGF- β' to 'TGF-β'. Please check all.
- Ref. 17 is regarding to polymer brushes instead of hydrogels. A recent review (Liu et al. Gels, 8(1), 46) related to zwitterionic hydrogels should be included to support such claim.
- Line 112-114, it is not clear that the contact angle results are not shown in the ms.
- The text in figure 2 is too small to be seen clearly.
- The title should be in a lower case.
Author Response
The research article by Gori et al. is well presented. All experiments necessary for indicating the effect of developing soft zwitterionic hydrogel coatings for reducing the foreign body reaction is well conducted and nicely presented through figures. However, there are some minor changes is to be done in order to improve the manuscript.
- We sincerely acknowledge the Reviewer for the interesting and useful suggestions that improve the quality of the manuscript and we made corrections accordingly, with the hope to have exhaustively addressed the Reviewer’s comments.
1. Typos. Line 78-79, 'Poly(ethylene glycol) (PEG)' to 'poly(ethylene glycol) (PEG)'; line 410, 'TGF- β' to 'TGF-β'. Please check all.
- All typos have been changed accordingly.
2. Ref. 17 is regarding to polymer brushes instead of hydrogels. A recent review (Liu et al. Gels, 8(1), 46) related to zwitterionic hydrogels should be included to support such claim.
- The suggested review article has been included in the manuscript.
3. Line 112-114, it is not clear that the contact angle results are not shown in the ms.
- We inserted the water contact angle results in the main text at page 4 (lines 148-151) with the corresponding photos and graph in the new Figure 1A and caption.
4. The text in figure 2 is too small to be seen clearly.
- Text font in Figure 2 has been enlarged.
5. The title should be in a lower case.
- The title has been converted to lower case.
Round 2
Reviewer 2 Report
The authors answered to the addressed queries and updated the document accordingly. Thank you.